# Opposite Effects of Work-Related Physical Activity and Leisure-Time Physical Activity on the Risk of Diabetes in Korean Adults

**DOI:** 10.3390/ijerph17165812

**Published:** 2020-08-11

**Authors:** Hyun Sook Oh

**Affiliations:** Department of Applied Statistics, College of Social Science, Gachon University, Seongnam 13120, Korea; hoh@gachon.ac.kr

**Keywords:** domestic and work-related physical activity, leisure-time physical activity, impaired fasting glucose(IFG), fasting blood glucose (FBG), Bayesian ordered probit model, diabetes

## Abstract

The object of this study was to examine the effects of domestic and work-related physical activity (DWPA) and leisure-time physical activity (LTPA) on the risk of diabetes, by categorizing fasting blood glucose (FBG) levels into normal, Impaired Fasting Glucose (IFG), and diabetes. The sample consisted of 4661 adults aged 30 years or above, and was chosen from the 2017 Korean National Health and Nutrition Examination Survey (KNHANES) data. Of all the subjects, 14.6% engaged in high-intensity DWPA and 6.25% in moderate-intensity DWPA; while 11.68% and 24.80% engaged in high- and moderate-intensity LTPA, respectively. The effects of both types of physical activities on the risk of diabetes were analyzed using a Bayesian ordered probit model. For those with high-intensity DWPA, the probability of the FBG level being normal was 5.10% (SE = 0.25) lower than for those with non-high-intensity DWPA, and the probabilities of IFG and diabetes were 3.30% (SE = 0.15) and 1.79% (SE = 0.09) higher, respectively. However, for those with high-intensity LTPA, the probability of the FBG level being normal was 2.54% (SE = 0.09) higher, and the probabilities of IFG and diabetes were 1.74% (SE = 0.07) and 0.80% (SE = 0.03) lower, respectively, than those with non-high-intensity LTPA. Likewise, for moderate-intensity DWPA and LTPA, the results were the same compared to low-intensity physical activities though the magnitude of the effects were smaller than for high-intensity. Thus, the activities related to work have a negative effect and those related to leisure have a positive effect. The criteria for physical activities to reduce the risk of diabetes should be set by separating these domains of physical activity, and new management strategies for diabetes are needed for people with moderate- or high-intensity DWPA.

## 1. Introduction

Exercise is effective not only for controlling blood sugar in diabetic patients but also for reducing cardiovascular risk, losing weight, protecting bone health, preventing metabolic diseases, and improving quality of life [1,2,3,4]. Many studies have reported that physical activity is effective in preventing and treating diabetes [5,6,7,8,9,10] and the US government and the World Health Organization(WHO) recommend adults to engage in at least 150 min of moderate-intensity physical activity, 75 min of vigorous-intensity physical activity, or an equivalent combination of the two intensities per week, for health, including diabetes prevention [11,12]. However, most of the previous studies on the effects of physical activity on the risk of diabetes have focused only on leisure-time or total (work-related and leisure-time) physical activity. Only a few studies have separated leisure-time and work-related physical activity, and have studied whether these physical activities affect the risk of diabetes differently and what levels of physical activity are needed to reduce the risk of type 2 diabetes [13,14,15,16].

Holterman et al. [14] established that there were opposing effects of occupational and leisure-time physical activity on global health, meaning that leisure-time physical activity is beneficial to health, but occupational physical activity at a moderate or high intensity level is rather harmful. Villegas et al. [16] showed that high-intensity physical activity in daily activities was inversely associated with high-intensity leisure-time physical activity and moderate- to high-intensity household activities were associated with an increased risk of type 2 diabetes. Therefore, it is necessary to examine the effects of other types of physical activity in addition to leisure-time activity on the risk of diabetes and compare them with the effects of leisure-time activity.

There were also inconsistent research results on the effects of LTPA intensity levels; a meta-analysis of cohort studies, mainly performed on Caucasians, suggested that the effect of reducing the risk of diabetes was more prominent with low-level leisure-time physical activity than with high-level [17]. However, it has been shown that vigorous-intensity exercise reduces the risk of diabetes and that lower-intensity exercise does not bring about any reduction in the risk of diabetes in Japanese workers [18]. Individuals performing moderate-intensity exercise might also perform vigorous-intensity exercise, and vice versa, so there is a possibility of mixed results due to a mixture of different intensity exercises [18]. The effects of exercise intensity levels on the risk of diabetes may also differ between Asians and Caucasians [19,20,21]. Therefore, it is necessary to investigate the effects of LTPA (including DWPA) intensity levels in reducing the risk of diabetes without mixing different intensity activities, while considering the race of the subjects of study.

Diagnosis of diabetes is based on FBG (FBG ≥ 126 mg/dL), current taking of antidiabetic medication(s), history of previous diabetes, or glycosylated hemoglobin (HbA1c ≥ 6.5%) [22,23]. FBG level is one of the most important measures by which the risk of diabetes can be predicted. In the case of diabetic patients, management of FBG is crucial because the incidence of complications increases with uncontrolled blood sugar [24]. FBG levels are generally classified into three categories, normal (FBG < 100), IFG (FBG:100–125), and diabetes (FBG ≥ 126) [22], with the risk of diabetes increasing in that order. IFG is known as the pre-diabetic stage, and it has been shown that IFG is a strong predictor of diabetes with high rate of conversion from IFG to diabetes in a cohort study in Taiwan [25].

Diabetes is the sixth leading cause of death in Korea, which has a high prevalence of diabetes, with about 13.7% of adults (≥30 years old) having diabetes and 24.8% of adults having IFG based on data from the KNHANES for 2013 to 2014 [23,26]. Therefore, diabetes is a major public health threat in Korea, and the potential risk of diabetes is very high due to the high rate of IFG. In this study, the association between the types of physical activity (DWPA and LTPA), taking into account intensity, and the risk of diabetes indicated by FBG level was investigated in Korean adults.

## 2. Methods

### 2.1. Data Collection

The KNHANES is a nationwide population-based survey of the health and nutrition status of Korean people conducted under the auspices of the Korea Centers for Disease Control and Prevention (KCDC) [27]. The KNHANES is composed of three parts with a total of about 800 questions: health interview, health examination, and nutrition survey. The health interview and health examination are conducted at mobile examination centers by trained specialists including physicians, medical technicians and health interviewers. The nutrition survey is conducted at the participant’s home by dieticians. Data are collected through self-report or face-to-face interview method, depending on the survey questionnaire type. The survey uses a complex, stratified sampling design in which sampling units are selected based on geographical area, sex, and age to select a representative sample of civilian, noninstitutionalized Korean people. The KNHANES provides statistics for health-related policies in Korea and also serves as a research infrastructure for studies on risk factors and diseases by supporting over 500 publications. KNHANES data are widely used by governmental organizations and researchers, and micro data is released on the KNHANES website (http://knhanes.cdc.go.kr).

This study used data from the 2017 KNHANES to investigate the association between types of physical activity and the risk of type 2 diabetes. Participants younger than 30 years were excluded because they were likely to have type 1 diabetes if they were younger than 30 years when diagnosed [28]. Therefore, of the 8127 individuals from the KNHANES (2017) data, 5727 people over the age of 30 which is the age to begin regular health checkups, and who were at an increased risk of developing type 2 diabetes, were studied. After excluding subjects with missing data on multiple variables, the remaining 4661 subjects were analyzed.

### 2.2. Assessment of DWPA and LTPA

The physical activity survey in the KNHANES (2017) uses the Korean version (K-GPAQ) of the Global Physical Activity Questionnaire (GPAQ) developed by the WHO and in use since 2014. It was verified that the K-GPAQ is a reliable and valid questionnaire to measure physical activity [29]. It is administered via face-to-face interviews, and consists of questions on physical activity related to work, leisure time, and movement between places over a week.

DWPA includes physical activity related to occupation, academic work, household chores, volunteer activities, athletic activities in school, agriculture, fishery, and stock farming. High-intensity DWPA is physical activity that keeps one breathless or one′s heartbeat very fast for at least 10 min, for instance, heavy lifting (about 20 kg or more), digging, ironworks, carrying objects up and down stairs, and physical activity in school such as soccer matches, basketball matches, swimming, and inline skating. Moderate-intensity DWPA is physical activity that keeps one slightly out of breath and one′s heartbeat slightly faster, for instance, cleaning, walking fast (less than 5.5 kmph), childcare, woodworking, caring for livestock, gymnastics, basketball practice and soccer practice.

LTPA includes physical activity related to sports, exercise, and leisure activities. High-intensity LTPA includes hiking, jumping rope, running (over 6.5 kmph), squash, and cycling (over 16 kmph), while moderate-intensity LTPA includes golf, bowling, gymnastics, fast walking (less than 5.5 kmph), slow running (less than 6.5 kmph), Pilates, other exercise (playing basketball, volleyball, swimming, etc.), dancing, and cycling (less than 16 kmph).

Subjects were first asked if they were involved in high-intensity DWPA (yes/no). If the answer was ‘no,’ they were asked about moderate-intensity DWPA. The same procedure was followed for high- and moderate-intensity LTPA, so there was no mixture of different intensity physical activities. Subjects who answered ′yes′ were asked the frequency of their physical activity (average days per week) and its duration (average hours and minutes per day). However, considering the very low response rate (less than 10%) for questions on frequency and duration, only the data on whether the subject engaged in physical activity was analyzed.

### 2.3. Other Variables

Socio-demographic factors including gender, age, education level, economic activity, marital status, household income level, and residential area; health behavioral factors including place-moving physical activity (PMPA), drinking, smoking, stress, sedentary time, and sleeping hours; and disease-related factors including hypertension, hypercholesterolemia, obesity, stroke, and heart disease, were controlled (Table 1). PMPA is the physical activity of walking or cycling for at least 10 min when moving between places, such as commuting to work, going to school, or going shopping.

Data for most socio-demographic factors and health behavioral factors were collected by self-reported questionnaires, while data pertaining to education, economic activity and physical activity were collected by face-to-face interviews. For disease-related factors, hypercholesterolemia, hypertension, and obesity were classified based on data collected by direct measurements. Hypertension was defined as systolic blood pressure ≥140 or diastolic blood pressure ≥90 mm Hg or taking antihypertensive medication(s); pre-hypertension stage was determined as 120–139/80–89 mm Hg, without meeting the criteria for hypertension. Hypercholesterolemia was defined as total serum cholesterol ≥200 mg/dL or taking pharmacologic lipid-lowering medication(s). Obesity was defined by body mass index(BMI) criteria and grouped into three categories: BMI < 25 (normal), BMI: 25–30 (overweight), and ≥30 (obesity). Ischemic heart disease was defined as being diagnosed with myocardial infarction or angina by a doctor, and stroke was defined as being diagnosed with a stroke by a doctor at the baseline medical examination.

The dependent variable, FBG, was measured indirectly by spectrophotometry after enzymatic reaction. Blood samples were collected through the veins in an empty stomach (for at least 8 h) and then refrigerated for FBG analysis. FBG levels were classified into three categories, normal (FBG < 100), IFG (FBG: 100–125), and diabetes (FBG ≥ 126).

### 2.4. Statistical Analysis

Descriptive statistics and chi-square tests were conducted as part of the preliminary analysis of the research data. All analyses were conducted using the statistical package R (version 3.6.1) (R Foundation for Statistical Computing, Vienna, Austria), and reflected the characteristics of the data collected through stratified two-stage cluster sampling. The dependent variable (FBG) was a categorical variable. To analyze its predictors, a Bayesian ordered probit model, which is applied when the dependent variable is an ordered categorical variable, was fitted.

Marginal effects are measured using the marginal probability of the dependent variable for a given explanatory variable by estimates of the regression coefficients for the unobserved latent variables in the model. The marginal effect of a specific variable is the difference in the marginal probability of the dependent variable caused by a change in the value of that variable, while other explanatory variables are controlled. For categorical variables, it is the value obtained by subtracting the marginal probability at a given level of the variable from the marginal probability at the reference level. For continuous variables, it is the change in marginal probability when the variable value increases by one unit for the other variables fixed, which represents instantaneous change. In this study, ordered categorical variables over four categories were treated as continuous variables. Proportional logistic regression for ordered categorical variables does not often fit the data as it requires that the assumption of proportional odds be satisfied, whereas the ordered probit model has no such requirement.

Gibbs sampling was used to implement the Bayesian method of ordered probit model analysis [30,31]. The Gibbs sampling algorithm is a simulation method that generates a joint probability sample of parameters using conditional distributions of parameters in complex models. The weight of each sample according to the stratified two-stage cluster sampling was applied in the algorithm, and the ′zelig′ function in the statistical package R (version 3.6.1) was used. All tests of significance were based on two-tailed probability at significance level 0.05.

## 3. Results

### 3.1. Characteristics of the Sample

Characteristics of the sample are summarized in Table 1. Out of all the subjects, 62.85% had normal FBG, 23.89% had IFG, and 8.27% were diabetic. The gender ratio of the sample was almost equal, while 84.61% of the subjects lived in urban areas, 91.27% were married, and 67.4% were economically active. The average age of the subjects was 51.54 years, the average monthly household income level (1 (below 10%) to 10 (top 10% or more)) was 6.73 (top 32.7%, about 5 million won), and the average education level (range: 1(unschooled) to 8(graduate level)) was 5.22, corresponding to high school graduate level.

With respect to health behavioral factors, 90.77% of the subjects had experience with drinking, 55.38% had never smoked, 2.13% had smoked fewer than five packs in their lifetime, and 42.49% had smoked more than five packs in their lifetime. Of all the subjects, 14.6% engaged in high-intensity DWPA and 6.25% in moderate-intensity DWPA; while 11.68% engaged in high-intensity LTPA, 24.80% in moderate-intensity LTPA, and 52.17% frequently engaged in PMPA. The average sedentary time per day was 7.91 h (SD = 3.55), the average sleeping time per day was 7 h (SD = 1.31), and the average level of stress experienced was (range: 1(weak) to 4(strong)) 2.16 (SD = 0.71).

With respect to disease-related factors, 1.94% of the subjects had experienced a stroke, 2.38% had ischemic heart disease (myocardial infarction or angina), 23.89% had hypercholesterolemia, 26.69% had prehypertension, 26.89% had hypertension, 30.61% had a Body Mass Index (BMI) between 25 and 30 (overweight), and 4.89% had a BMI of 30 or higher (obese).

### 3.2. Clinical Diagnosis of Diabetes

The relationship between FBG levels and diagnoses of diabetes from the clinicians was analyzed (Table 2). It was found that 1.26% were diagnosed with diabetes among subjects with normal FBG level, 9.46% among the subjects with IFG, and 61.55% among subjects being diabetic (*p* < 0.001).

### 3.3. Factors Affecting the Risk of Diabetes

We applied the Bayesian ordered probit model with the Gibbs sample algorithm to generate a joint probability sample of parameters. After generating a total of 255,000 samples, except for the initial (burn-in time) 5000, one in five was extracted to secure sample independence, making the final sample size 50,000. The convergence of the algorithm was confirmed using the path diagrams of the sample and Heidelberger and Welch’s convergence diagnostic [32,33]. Path diagrams for a few variables only are shown in Figure 1 since the path diagrams for all variables would take up too much space. In the Heidelberger and Welch’s convergence test for the null hypothesis that the sampled values came from a stationary distribution, the null hypothesis was accepted because all *p*-values for each variable were greater than 0.05 (not shown here). Post comparison, the actual frequencies and the estimated frequencies in the applied Bayesian ordered probit model were found to be very close (Table 3).

The results of the analysis are summarized in Table 4. The significance of the variable was judged by the coefficient β value, where, if β > 0, the probability that the FBG level indicates IFG or diabetes increased as the value of the variable increased. Conversely, if β < 0, the probability that the FBG level indicates IFG or diabetes decreased as the value of the variable increased. 

All socio-demographic variables were found to be significant predictors of FBG levels. Gender and education level had β < 0, and residential area, marital status, economic activity, age, and household income had β > 0. The marginal effects for females were 11.78% for normal and –6.21% and −5.63% for IFG and diabetes, respectively. In other words, the probability of FBG being normal was 11.78% higher in females than in males, and the probability of IFG and diabetes in females was 6.21% and 5.63% lower, respectively, than in males. The marginal effects for residential area (rural) were −0.4% for normal FBG and 0.27% and 0.13% for IFG and diabetes, respectively. The marginal effects for married individuals were −8.55% for normal FBG and 6.08% and 2.47% for IFG and diabetes, respectively. The marginal effects for economic activity (yes) were −2.75% for normal FBG, 1.87% for IFG, and 0.88% for diabetes. The marginal effects for age were −0.62% for normal FBG, 0.42% for IFG, and 0.20% for diabetes. The marginal effects for household income were −0.53% for normal FBG, 0.36% for IFG, and 0.17% for diabetes. The marginal effects for education level were 2.22% for normal FBG, −1.50% for IFG, and −0.72% for diabetes.

All of the health behavioral variables except for sedentary time were found to be significant. Among the significant variables, those with β < 0 were drinking, PMPA, high-intensity LTPA, moderate-intensity LTPA, and sleeping time; while those with β > 0 were smoking, high-intensity DWPA, moderate-intensity DWPA, and stress.

The marginal effects for drinking (yes) were 4.6% for normal FBG, –2.8% for IFG, and –1.8% for diabetes; those for smoking (<5 packs) were −9.97% for normal FBG, 6.19% for IFG, and 3.78% for diabetes; and those for smoking (≥5 packs) were −6.11% for normal FBG, 4.08% for IFG, and 2.03% for diabetes. The marginal effects for high-intensity DWPA (yes) were −5.10% for normal FBG, 3.30% for IFG, and 1.79% for diabetes; while those for moderate-intensity DWPA (yes) were −1.18% for normal FBG, 0.79% for IFG, and 0.39% for diabetes. The marginal effects for PMPA (yes) were 1.71% for normal FBG, −1.15% for IFG, and −0.55% for diabetes. The marginal effects for high-intensity LTPA (yes) were 2.54% for normal FBG, −1.74% for IFG, and −0.80% for diabetes; while those for moderate -intensity LTPA (yes) were 0.59% for normal FBG, −0.40% for IFG, and −0.19% for diabetes. The marginal effects for sleeping time were 0.53% for normal FBG, –0.36% for IFG, and –0.17% for diabetes; and the marginal effects for stress were –0.50% for normal FBG, 0.34% for IFG, and 0.16% for diabetes.

All of the disease-related variables were found to be significant with β > 0. The marginal effects for stroke (yes) were −15.89% for normal FBG, 9.29% for IFG, and diabetes 6.60%; those for heart disease (yes) were −6.23% for normal FBG, 4.01% for IFG, and 2.22% for diabetes, and those for hypercholesterolemia (yes) were −8.77% for normal FBG, 5.71% for IFG, and 3.06% for diabetes. For prehypertension (yes), the marginal effects were −8.64% for normal FBG, 5.63% for IFG, and 3.01% for diabetes; while for hypertension, they were −16.73% for normal FBG, 10.72% for IFG, and 6.01% for diabetes. For a BMI indicating overweight, the marginal effects were −12.36% for normal FBG, 7.99% for IFG, and 4.37% for diabetes; while for a BMI indicating obesity, they were −28.42% for normal FBG, 14.22% for IFG, and 14.20% for diabetes.

## 4. Discussion

This study analyzed the relationship between the risk of diabetes as indicated by FBG level and two types of physical activity, DWPA and LTPA, in adults over 30 years of age, using data of Korean citizens collected through KNHANES (2017) at the national level. For each of DWPA and LTPA, it was classified as high, moderate, and low intensity level, without a mixture of different intensity physical activities. However, quantification considering the frequency and duration of physical activity was not possible due to the large number of missing values.

The percentage of people with IFG and diabetes was 28.89% and 8.27%, respectively. Based on the KNHANES (2013–2014) data, Won et al. [23] reported that 24.8% of adults over 30 years of age had IFG, indicating that the rate of IFG increased by about 4%. Glycosylated hemoglobin, family history of diabetes, and symptoms of diabetes are considered for a diagnosis of diabetes in Korea [34] even if the FBG level of the individual is low, so the actual rate of people with diabetes would be higher than 8.27%. In the sample for this study, 1.26% were diagnosed with diabetes for the subjects with normal FBG level, 9.46% for the subjects with IFG, and the total proportion of people diagnosed with diabetes by clinicians was 10.32%.

The main finding of this study is that LTPA and DWPA have opposite effects on the risk of diabetes. Subjects with high- or moderate-intensity DWPA had greater probabilities of IFG and diabetes than the low-intensity DWPA group. However, the high- or moderate-intensity LTPA group had lower probabilities of IFG and diabetes than the low-intensity LTPA group. In particular, the magnitude of the effect was much greater when the intensity of activity was high than when it was moderate for both DWPA and LTPA. Thus, physical activity had a negative effect when related to work, and a positive effect when related to leisure. The effect was noticeable in case of high-intensity physical activity.

The findings of this study related to LTPA coincide with those of most previous studies that have established an inverse association between diabetes and LTPA. However, Aune et al. [17] reported from a meta-analysis of several cohort studies that the effect of reducing the risk of diabetes was more prominent with low-level LTPA than with high-level LTPA. Conversely, Honda et al. [18], in a cohort study of Japanese workers, found that vigorous-intensity leisure-time exercise reduces the risk of diabetes and that lower-intensity leisure-time exercise does not bring about any reduction in the risk of diabetes. Steinbrecher et al. [35] established that low- or moderate-intensity leisure-time physical activity (sports) may have little or no benefit for non-Caucasian groups with regard to the risk of diabetes by pointing out that most of the studies showing such results were performed with Caucasian subjects. Combining these findings with the results of the present study has made it clearer that high-intensity LTPA is effective in reducing the risk of diabetes in Asians, but low-intensity LTPA is ineffective.

However, the results of this study related to DWPA deviated from those of previous studies. Most of the previous studies have reported positive or no effect of work-related physical activity in reducing the risk of diabetes [17,18,35,36,37]. In several studies, work activity was combined with leisure-time activity and defined as total physical activity, or it was separated from leisure-time activity but was limited to occupation-related activity [18,36].

In the present study, DWPA consisted of physical activity in all areas of life, including work, family, and school, except LTPA and PMPA [35,38,39]. Villegas et al. [16] showed that while occupational physical activity, classified using job codes, was not associated with the development of diabetes, moderate- to high-intensity household activities were associated with an increased risk of type 2 diabetes in the Shanghai women’s health study. Since the intensity of physical activity could vary in the same workplace depending on the type of work, using job codes to evaluate this seems questionable. Nevertheless, it is worth noting that moderate- to high-intensity household activities increased the risk of diabetes. Interestingly, Ku et al. [39] found that low levels of DWPA were associated with a lower risk of mortality in patients with cardiovascular disease, while moderate and high levels were not. Holterman et al. [14] also established that occupational physical activities of moderate and high intensity increase the risk of long-term sickness absence.

Since DWPA is more likely to be obligatory, repetitive or routine, higher levels of DWPA may be excessively demanding [39,40]. This fact could explain the findings of this study that moderate- and high-intensity DWPA increases the risk of diabetes. Earlier, most studies on risk factors for diabetes focused on LTPA, resulting in physical activity being recognized as a key strategy for preventing diabetes. However, results of the present study suggest that a new diabetes management system is needed for people with moderate- or high-intensity DWPA. In fact, high-intensity physical activity in daily activities such as walking, climbing stairs, cycling, and household activities is shown to be inversely associated with high-intensity LTPA [16].

Exercise for 40 to 60 min per day at moderate or high intensity in guidelines for diabetes in Korea is recommended [34]. However, this recommended exercise is mainly focused on moderate-intensity daily-life activities and leisure-time exercises, and there is no consideration of work-related physical activities. Hence, more emphasis is needed on high-intensity exercise, and new physical activity strategies considering DWPA and LTPA are needed, for which further research on the risk of diabetes focusing on DWPA is necessary.

In addition to LTPA and DWPA, PMPA was found to be effective in reducing the risk of diabetes. This finding was consistent with previous studies, which showed that physical activity related to walking or commuting had an inverse relationship with the risk of diabetes [36]. Meanwhile, sedentary time was not found to have a significant effect in this study. This may be explained by the fact that the study only measured the total time spent sitting per day. More comprehensive measurement of sedentary time may alter the results.

Out of the demographic factors, gender, residential area, marital status, economic activity, age, household income, and education level were found to be significantly associated with the risk of diabetes. Consistent with the results of previous studies, it was found that males, older individuals, individuals with higher income, and individuals with a lower education level had a higher risk of diabetes [41]. Unmarried people were found to have a lower risk of diabetes than married people. This may be because the average age of unmarried subjects (39 years; SD = 0.52) was lower compared to the average age of married subjects (53 years; SD = 0.46) in this study.

Among health behaviors, the drinking group had a lower risk of diabetes than the non-drinking group when the only consideration was the presence of drinking. However, when the frequency and amount of drinking were considered, a positive correlation between drinking and FBG levels was found [42,43]. Smokers, people with higher stress levels, and those with shorter sleeping times were found to have a higher risk of diabetes.

Chronic diseases such as stroke, ischemic heart disease, hypercholesterolemia, hypertension, and obesity were more positively associated with an increased risk of diabetes than socio-demographic and health behavioral factors. In particular, obesity had the strongest association with an increased risk of diabetes.

As this was a cross-sectional study, a direct causal relationship between variables could not be identified. Cohort studies tend to be more suitable for causal analysis, but may also pose serious difficulties in controlling causal variables. For example, when a group is classified by the level of physical activity of its members and their FBG levels are measured after several years of observation, it is almost impossible that the initially classified group would retain the same characteristics over that time. Post-assessment of FBG levels and physical activity (including in-person interviews), the follow-up period in these studies is usually set for one to two years, as it is difficult to accurately remember or express changes in physical activity patterns beyond that point. With a short follow-up period, the cohort study would differ only slightly from a cross-sectional study.

The present study did not consider the frequency and duration of physical activity owing to the response rate being very low (less than 10%). In addition, there was no mixture of different intensities within each physical activity, but the possibility of mixing DWPA and LTPA existed. In other words, subjects with moderate- or high-intensity DWPA may be the same as those with moderate- or high-intensity LTPA. In the future, we intend to analyze physical activity using quantitative evaluation criteria without a mixture of DWPA and LTPA, as well as different intensities within each physical activity. Another limitation is that diabetes-related factors such as family history of diabetes, kidney disease, and triglyceride levels were not included in the control variables because more than 1000 values were missing in the data. Moreover, since gender differences have been found in the risk and prevention of type 2 diabetes and the effects of physical activity such as weight loss [44,45], it would have been more appropriate to analyze the data separately by gender, but our sample size was not large enough. This will be studied in the future.

## 5. Conclusions

Moderate- or high-intensity LTPA reduces the risk of diabetes. This applies especially to Asians, including Koreans, and not to Caucasians. Conversely, moderate- or high-intensity DWPA increases the risk of diabetes, and the risk is higher for activities of high intensity. Therefore, even if the activities are of equally moderate or high intensity, activities related to work have a negative effect and those related to leisure have a positive effect. Therefore, we recommend that the criteria for physical activities in relation to the risk of diabetes should be set by separating the domains of physical activity. Further research on the risk of diabetes with a focus on DWPA is necessary, and new management strategies for diabetes are needed for people with moderate- or high-intensity DWPA.

## Figures and Tables

**Figure 1 ijerph-17-05812-f001:**
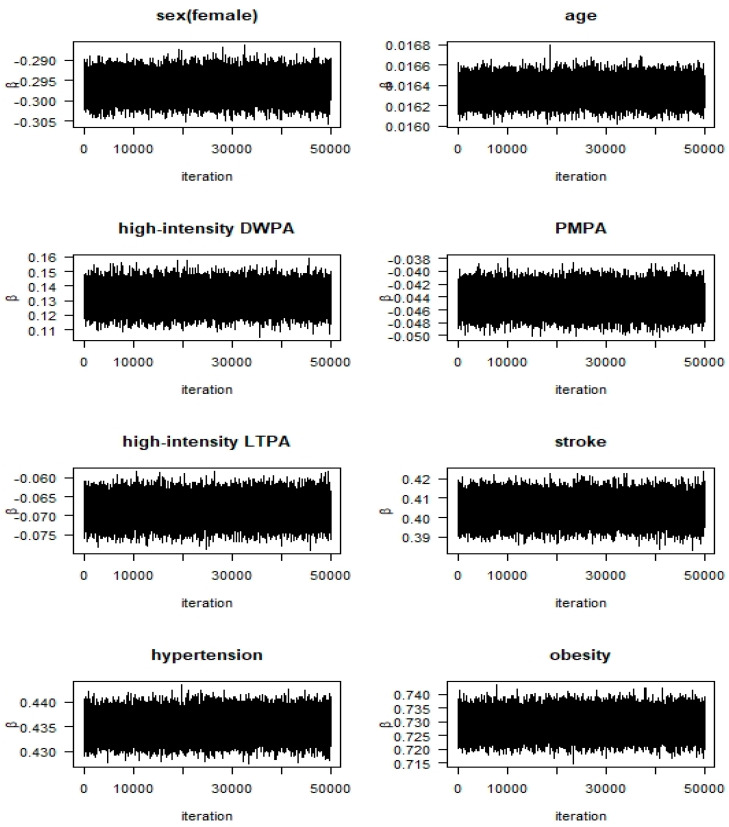
Path diagrams for variables.

**Table 1 ijerph-17-05812-t001:** Sample Characteristics.

Factors	Variables	Levels	*N*	wtd %	Mean (sd)	wtd Mean (sd)
FBG		Normal (<100)	2837	62.85		
		FBG IFG (100–125)	1385	28.89		
		Diabetes (≥126)	439	8.27		
socio- demographic factors	sex	male	2069	49.52		
		female	2562	50.48		
	residence	urban	3793	84.61		
		rural	868	15.39		
	marital status	unmarried	322	8.73		
		married	4339	91.27		
	occupation	no	1744	32.6		
		yes	2917	67.4		
	age	30~80(year)			54.66 (13.83)	51.54 (13.26)
	household income (monthly)	1–10 1: <10%–10: >90%			6.31 (3.20)	6.73 (3.06)
	education	1–8 1:unschooled– 8:graduate level			5.22 (1.63)	5.47 (1.56)
health behaviors	drinking	never	523	9.23		
		yes	4138	90.77		
	smoking	never	2761	55.38		
		<5 pack	89	2.13		
		≥5 pack	1811	42.49		
	high-intensity DWPA	no	4614	98.54		
		yes	47	14.6		
	moderate-intensity DWPA	no	4402	93.75		
		yes	259	6.25		
	PMPA	no	2209	47.73		
		yes	2452	52.17		
	high-intensity LTPA	no	4217	88.32		
		yes	444	11.68		
	moderate-intensity LTPA	no	3588	75.20		
		yes	1078	24.53		
	sedentary time (hour per day)				7.92 (3.53)	7.91 (3.55)
	sleeping time (hour per day)				7.02 (1.36)	7.00 (1.31)
	stress (1: hardly– 4: very much)				2.13 (0.72)	2.16 (0.71)
chronic diseases	stroke	no	4546	98.06		
		yes	115	1.94		
	heart disease	no	4515	97.62		
		yes	146	2.38		
	hypercholesterolemia	no	3451	76.11		
		yes	1210	23.89		
	hypertension	normal	1873	43.42		
		pre-hypertension	1197	26.69		
		hypertension	1591	29.89		
	obesity	normal(BMI < 25)	3004	64.50		
		overweight (BMI25–30)	1427	30.61		
		obesity (BMI ≥ 30)	230	4.89		

Abbreviations: wtd = weighted; sd = standard deviation; FBG = fasting blood glucose; IFG = impaired fasting glucose; DWPA = domestic and work-related physical activity; LTPA = leisure-time physical activity; PMPA = place-moving physical activity; BMI = body mass index.

**Table 2 ijerph-17-05812-t002:** Fasting blood glucose (FBG) and diabetes diagnosis by clinicians.

FBG	Diabetes Diagnosis		Wald F †	*p*-Value
	Yes (wtd%)	No (wtd%)		
normal	48(1.26)	2789(98.74)	111.27	0.000
IFG	153(9.46)	1232(90.54)		
diabetes	280(61.55)	159(38.45)		

†, adjusted Wald F test for complex sample. Abbreviations: FBG = fasting blood glucose; IFG = impaired fasting glucose.

**Table 3 ijerph-17-05812-t003:** Relative frequency of each FBG category.

Frequency	Normal	IFG	Diabetes
observed frequency(weighted)	0.6285	0.2889	0.0827
estimated frequency	0.6059	0.3009	0.0933

Abbreviations: FBG = fasting blood glucose; IFG = impaired fasting glucose.

**Table 4 ijerph-17-05812-t004:** Estimates of marginal effects.

Factors	Variables	$\hat{\beta }$ (*SE*)	Marginal Effects		
			Normal (*SE*)	IFG (*SE*)	Diabetes (*SE*)
Demographic factors	sex (female)	−0.2963 (0.0023)	0.1178 (0.0009)	−0.0621 (0.0003)	−0.0563 (0.0006)
	residence (rural)	0.0105 (0.0021)	−0.0040 (0.0008)	0.0027 (0.0005)	0.0013 (0.0003)
	marital status (married)	0.2340 (0.0031)	−0.0855 (0.0011)	0.0608 (0.0008)	0.0247 (0.0003)
	occupation (yes)	0.0727 (0.0018)	−0.0275 (0.0007)	0.0187 (0.0005)	0.0088 (0.0002)
	age	0.0164 (0.0000)	−0.0062 (0.0000)	0.0042 (0.0000)	0.0020 (0.0000)
	household income	0.0140 (0.0003)	−0.0053 (0.0001)	0.0036 (0.0001)	0.0017 (0.0000)
	education	−0.0585 (0.0006)	0.0222 (0.0002)	−0.0150 (0.0002)	−0.0072 (0.0001)
health behaviors	Drinking (yes)	−0.1163 (0.0026)	0.0458 (0.0010)	−0.0277 (0.0005)	−0.0181 (0.0005)
	Smoking (<5pack)	0.2553 (0.0052)	−0.0997 (0.0021)	0.0619 (0.0012)	0.0378 (0.0009)
	Smoking (≥5pack)	0.1601 (0.0021)	−0.0611 (0.0008)	0.0408 (0.0005)	0.0203 (0.0003)
	high-intensity DWPA (yes)	0.1318 (0.0063)	−0.0510 (0.0025)	0.0330 (0.0015)	0.0179 (0.0009)
	moderate-intensity DWPA (yes)	0.0309 (0.0032)	−0.0118 (0.0012)	0.0079 (0.0008)	0.0039 (0.0004)
	PMPA (yes)	−0.0449 (0.0015)	0.0171 (0.0006)	−0.0115 (0.0004)	−0.0055 (0.0002)
	high-intensity LTPA (yes)	−0.0675 (0.0025)	0.0254 (0.0009)	−0.0174 (0.0007)	−0.0080 (0.0003)
	moderate-intensity LTPA (yes)	−0.0156 (0.0019)	0.0059 (0.0007)	−0.0040 (0.0005)	−0.0019 (0.0002)
	sedentary time	0.0004 (0.0002)	−0.0002 (0.0001)	0.0001 (0.0001)	0.0000 (0.0000)
	sleeping time	−0.0140 (0.0006)	0.0053 (0.0002)	−0.0036 (0.0001)	−0.0017 (0.0001)
	stress	0.0133 (0.0011)	−0.0050 (0.0004)	0.0034 (0.0003)	0.0016 (0.0001)
diseases	Stroke (yes)	0.4039 (0.0049)	−0.1589 (0.0019)	0.0929 (0.0009)	0.0660 (0.0010)
	heart disease (yes)	0.1608 (0.0044)	−0.0623 (0.0017)	0.0401 (0.0011)	0.0222 (0.0007)
	hypercholesterolemia (yes)	0.2279 (0.0017)	−0.0877 (0.0007)	0.0571 (0.0004)	0.0306 (0.0003)
	pre-hypertension (yes)	0.2246 (0.0019)	−0.0864 (0.0007)	0.0563 (0.0005)	0.0301 (0.0003)
	Hypertension (yes)	0.4354 (0.0020)	−0.1673 (0.0008)	0.1072 (0.0005)	0.0601 (0.0003)
	Overweight (yes)	0.3212 (0.0016)	−0.1236 (0.0006)	0.0799 (0.0004)	0.0437 (0.0002)
	Obesity (yes)	0.7281 (0.0033)	−0.2842 (0.0012)	0.1422 (0.0004)	0.1420 (0.0009)

Abbreviations: *SE* = standard error; IFG = impaired fasting glucose; DWPA = domestic and work-related physical activity; LTPA = leisure-time physical activity; PMPA= place-moving physical activity.

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
