# Peer review of "Opposite Effects of Work-Related Physical Activity and Leisure-Time Physical Activity on the Risk of Diabetes in Korean Adults"

_ijerph, 2020, doi:10.3390/ijerph17165812_

Round 1
Reviewer 1 Report
Dear author
I have read your manuscript and I consider it to be interesting work. I would like to propose it for publication, but first you have to make some changes.
Abstract: This section may start with the aim of the study.
Introduction: In this section I consider necessary indicate the incidence of diabetes in the Korean population
Methods: More information about KNHANES questionnaire is required. How many questions did it has? Has it been previously used in other studies?
A practical applications section geared towards Korean Government policies is necessary.
Reviewer 2 Report
I find this work very interesting because provide further evidence about the benefits of leisure-time PA on the risk of diabetes and that high-intensity domestic, work-related PA might be not positive for reducing the risk of diabetes. The work is in general very well written. However, I think it might be improved. I hope my comments below help to this purpose.
INTRODUCTION
Page 1, Line 34-35: I understand this sentence. However, somehow is against the findings to this study and the rationale provided in the second paragraph. I.e. when the authors state that “high-intensity household activities were associated with an increased risk of type 2 diabetes (line 43)”. Therefore, I suggest deleting this sentence or rewriting it in order to better match with the next paragraph.
Page 2, Line 3-8: I am not totally happy on how this paragraph matches with the introduction. I think the ideas discussed in this paragraph do not complement the previous ones” (line 7).
- I think it explores some discrepancies about the effectiveness of Leisure-time PA on the risk of diabetes. Therefore, I suggest rewriting to emphasise this fact
- I think the sentence “So it is useful to see why this difference has occurred” is unfortunate as the authors do not study the discrepancies between the findings from Aune et al. and Honda et al. and how they occurred.
Page 2. I would merge the last two paragraphs of the introduction.
METHODS
- I think that further information is required to know how sociodemographic factors were collected. Same for health behavioural factors and disease-related factors.
- Same for the information for FBG levels. How they were collected?
Data collection
Page 2, Lines 30-33: I think a reference is required to justify why only people over 30 were selected.
Statistical analysis
I recommend merging paragraph 1 and 2
RESULTS
Table 1: I think the authors should add the meaning of all the abbreviation to the table foot.
Table 2: further explanation of how FBG was collected is required to understand this table.
DISCUSSION
In general, I am quite happy with the discussion. However, I think that the discussion should be conducted considering the main limitation of this study. I mean that when the authors are comparing their outcomes with the previous ones, they should inform that differences with previous studies might be due to the fact this study did not consider the frequency and duration of PA. I understand that limitations usually go at the end, but in this case, I think this limitation is very significant. Maybe it is enough if the authors inform about this limitation at the beginning of the discussion
Reviewer 3 Report
In general the study seems to have a lack in bibliographic support.
Introduction
- I suggest to the authors to improve this section developing the link between diabetes and PA, especially in the fourth paragraph where the concept of FBG is poorly explored.
- Maybe the coma should be removed in the following expression “domestic, work-related physical activity“
Methods
I would like to suggest to authors to separate the sample by sexes, as the probability to have diabetes is not the same for both sexes.
Discussion
“high-intensity LTPA proved effective, moderate-intensity LTPA was slightly effective, and low- intensity LTPA was not effective in reducing the risk of diabetes among Asians.” – Could the authors give some insights of that phenomenon?
The subjects that showed moderate- or high-intensity DWPA were the same that showed higher LTPA? Could the first be the “excuse” for not having a higher LTPA?
Reviewer 4 Report
This study analyzed the effects of domestic, work-related physical activity (DWPA) and 10 leisure-time physical activity (LTPA) on the risk of diabetes, by categorizing fasting blood glucose 11 (FBG) levels into the normal level, Impaired Fasting Glucose (IFG), and diabetes in Korean adults.
Minor corrections:
Title: a "the" must be added before "risk".
Introduction: Must be improved. More references are needed about previous studies on the effects of physical activity on diabetes, and what kind of exercise programs have been developed in this way.
Line 14 to 20: must be moved to the methods section.
The authors must finish the introduction section with a clear research objective.
Results: In table 1, the "comma" in N column is unclear. If the authors are describing "thousands", it is better without the comma. For example, in FBG, change 2,837 by 2837. Revise all the data.
